# Gravitational multipoles in general stationary spacetimes

**Daniel R. Mayerson⋆**

Institute for Theoretical Physics, KU Leuven, Celestijnenlaan 200D, B-3001 Leuven, Belgium

⋆ daniel.mayerson@kuleuven.be

## Abstract

The Geroch-Hansen and Thorne (ACMC) formalisms give rigorous and equivalent definitions for gravitational multipoles in stationary vacuum spacetimes. However, despite their ubiquitous use in gravitational physics, it has not been shown that these formalisms can be generalized to non-vacuum stationary solutions, except in a few special cases. This paper shows how the Geroch-Hansen formalism can be generalized to arbitrary non-vacuum stationary spacetimes for metrics that are sufficiently smooth at infinity. The key is the construction of an improved twist vector, which is well-defined under a mild topological condition on the spacetime (which is automatically satisfied for black holes). Ambiguities in the construction of this improved twist vector are discussed and fixed by imposing natural "gauge fixing" conditions, which also immediately lead to the equivalence between the Geroch-Hansen and Thorne formalisms for such arbitrary stationary spacetimes.



# 1 Introduction and summary

Multipole moments of a field encode the angular structure of the field as determined by its sources; successive multipole moments can typically be read off from the angular dependence of terms in an asymptotic radial expansion. This makes it a tricky business to define gravitational multipoles in general relativity due to general coordinate invariance. Nevertheless, Geroch [1, 2] and Hansen [3] managed to define and calculate gravitational multipoles of stationary, asymptotically flat vacuum spacetimes in an elegant, manifestly coordinate-invariant formalism. Thorne [4] developed a separate formalism for extracting coordinate-independent multipoles from a stationary, vacuum metric, which relies on properties of a preferred family of coordinate systems called asymptotically Cartesian and mass-centered (ACMC) coordinates. These two formalisms were shown to give equivalent definitions of multipoles by Gürsel [5].

There are two families of gravitational multipole tensors in a stationary, vacuum spacetime: the mass multipoles $M_{a_1 \cdots a_\ell}$ and the current multipoles (or angular momentum multipoles) $S_{a_1 \cdots a_\ell}$. For example, for Kerr, the multipole tensors reduce to single numbers $M_\ell, S_\ell$ at each order due to axisymmetry, and are given by $M_\ell = M(-a^2)^\ell$ and $S_\ell = Ma(-a^2)^\ell$. The most familiar multipoles are the mass $M_0 = M$ and angular momentum $S_1 = Ma$.

The original Geroch-Hansen and Thorne formalisms, as well as the proof of their equivalence, are all stated in asymptotically flat vacuum spacetimes.[1] The Geroch-Hansen formalism was generalized to electrovacuum solutions (i.e. solutions to the Einstein-Maxwell equations) [7–9], and scalar-tensor theories [10]. However, for a generic matter content, it has never been shown how — or if — the Geroch-Hansen formalism can be generalized. In addition, the equivalence between the Geroch-Hansen and Thorne formalisms has not been generalized beyond vacuum spacetimes.

Since the extension of Geroch-Hansen to stationary spacetimes with arbitrary matter content was uncertain, it is also not clear if the entire concept of the two families of multipole tensors $M_{a_1 \cdots a_\ell}$ and $S_{a_1 \cdots a_\ell}$ still applies in such theories. For example, it was suggested that there could exist theories in which stationary solutions admit a third family of multipole tensors [11].[2]

---

[1]The condition of asymptotic flatness can be relaxed to include a NUT parameter [6].

[2]Of course, it is clear that in the presence of matter, more information is needed besides these two metric multipole families for the full metric reconstruction; a simple illustration of this is the Kerr and Kerr-Newman black holes which both have identical gravitational multipoles $M_{a_1 \cdots a_\ell}$ and $S_{a_1 \cdots a_\ell}$. Only in vacuum are the two

Nevertheless, the usage of the multipoles $M_{a_1 \cdots a_\ell}$ and $S_{a_1 \cdots a_\ell}$ are ubiquitous in modern gravitational physics, and especially in phenomenology. For example, multipoles have been discussed for solutions of $\mathcal{N} = 2$ supergravity theories (with many scalars and gauge fields) [15–19], or solutions of higher-derivative gravity theories [20–22]. Multipoles have been argued to be important, theory-independent characteristics of the metric in gravitational wave phenomenology, as pioneered by Ryan [23, 24] and developed and generalized in other works [25–28]. Measuring the mass quadrupole is already an important facet of tests of deviations from general relativity with current gravitational wave detections, see e.g. [29].

It is clear that there is a need to expand our formal definitions and understanding of gravitational multipoles beyond (electro)vacuum spacetimes, to include solutions in theories with arbitrary matter content. That is precisely the goal of this paper.

## 1.1 Summary

This paper discusses how the Geroch-Hansen formalism can be generalized to stationary spacetimes with arbitrary matter Lagrangians, in order to give a rigorous definition of gravitational multipoles for any stationary spacetime. In addition, I show that the equivalence between the Geroch-Hansen and Thorne multipole formalisms similarly generalizes to arbitrary stationary spacetimes. The key to these proofs is the definition and properties of an "improved" twist vector $\omega_\mu^I$ in the Geroch-Hansen formalism (Section 3.1), which can be formulated in terms of the energy-momentum tensor featuring in Einstein's equations.

For generic matter content, this paper shows (Section 3.2) that this improved twist vector can be mathematically defined as long as a certain *closed* two-form constructed from the energy-momentum tensor on constant-time slices of the spacetime is also an *exact* form. Physically, this condition is that there are no spatial two-cycles over which there is a net flux of perpendicular (radial) momentum. This is a rather reasonable assumption for stationary spacetimes. For example, for a single, stationary black hole with a spherical horizon, asymptotic flatness ensures this condition is always satisfied so that the construction of $\omega_\mu^I$ is always guaranteed to hold.

The construction of the improved twist vector further contains an ambiguity or "gauge dependence" which affects the (current) multipole moments. I discuss (Section 3.3) the "gauge fixing" conditions that must be imposed upon the improved twist vector in order for the multipole moments to be unambiguously defined. With the proposed "gauge fixing" conditions, the equivalence of the Geroch-Hansen and Thorne multipoles also immediately follows for such arbitrary spacetimes.

Along the way, I also exhibit (Section 3.1.1) the explicit form of the improved twist vector $\omega_\mu^I$ for $\mathcal{N} = 2$ supergravity with an arbitrary number of vector multiplets, which has not been reported previously.

An assumption that must be made on the metric, beyond asymptotic flatness, is sufficient smoothness at infinity (see Sections 2.1 and 2.3 for a more precise definition and discussion) and the existence of an ACMC coordinate frame (see Section 2.2). Whereas these assumptions seem rather intuitively mild, they do leave open the possibility that the Thorne ACMC formalism is slightly more general, in the sense that metrics could exist where the Thorne ACMC formalism applies but the Geroch-Hansen formalism does not. (This is discussed further in Sections 3.4.3 and 3.4.4.)

The generalization presented here of Geroch-Hansen and Thorne to non-vacuum stationary spacetimes puts the definition of multipole moments and their usage in characterizing arbitrary stationary spacetimes on firmer mathematical ground. These multipoles being well-defined has been implicitly assumed in many (recent) works, but this paper is the first to show

---

multipole families sufficient information to reconstruct the metric unambiguously [12–14].

that gravitational multipoles should indeed be unambiguous observables in the presence of arbitrary matter fields.

An earlier partial discussion of generalizing the Geroch-Hansen formalism beyond vacuum spacetimes was given in [30] in the context of so-called bumpy black holes. There — as is also of the essence in this paper — the key fact allowing the generalization of the multipole formalism was the sufficient fall-off of the energy-momentum tensor; however, for many such bumpy black holes, only the first few multipole moments can be defined as the (improved) twist vector cannot be defined at higher orders in $1/r$. Correspondingly, the metric cannot be brought into ACMC form beyond a certain order.[3] Other attempts to generalize the notion of multipoles include generalizing multipoles to de Sitter spacetimes [31] using the Noether charge formalism of [11] which allows generalization of multipoles to non-stationary spacetimes. Perhaps the most obvious direction for future work is to understand how also the Noether charge formalism of [11] generalizes to general non-vacuum spacetimes, and its relation with the the (extended) Geroch-Hansen and Thorne formalisms discussed here.

Section 2 reviews the Geroch-Hansen and Thorne formalisms and the proof of their equivalence, all for vacuum stationary spacetimes. In Section 3, I discuss the generalization to non-vacuum spacetimes of the Geroch-Hansen formalism and multipole definitions by constructing the improved twist vector and discussing its properties, and I discuss the necessity of the assumptions of the existence of an ACMC coordinate frame and the smoothness of the metric at infinity.

## 2 Geroch-Hansen and Thorne formalisms in vacuum

This Section is a review of the formalisms of Geroch-Hansen [2,3] and Thorne [4] which both define the gravitational multipoles $M_{a_1 \cdots a_\ell}$ and $S_{a_1 \cdots a_\ell}$ of a stationary, vacuum spacetime, as well as the proof of their equivalence by Gürsel [5].

### 2.1 Geroch-Hansen multipoles

An elegant and manifestly coordinate-invariant formalism for defining multipoles of a vacuum spacetime was developed by Geroch [2] for static spacetimes, and later expanded to stationary spacetimes by Hansen [3]. (See also [32].)

Let $\xi$ be the (asymptotically) timelike Killing vector of the four-dimensional stationary, asymptotically flat spacetime with metric $g_{\mu\nu}$; the scalar field $\lambda$ is given by its norm:

$$\lambda = \xi^2. \tag{1}$$

We can define a manifold $\overline{\mathcal{M}}$ as the collection of the orbits of $\xi$ — this is indeed a (three-dimensional, Riemannian) manifold [33,34] and a natural metric on it is:

$$\overline{h}_{\alpha\beta} = g_{\alpha\beta} - \lambda^{-1}\xi_\alpha \xi_\beta. \tag{2}$$

The metric $\overline{h}_{\alpha\beta}$ can also be used to project tensors from the four-dimensional spacetime to $\overline{\mathcal{M}}$; note that $\xi^\mu \overline{h}^\alpha_\mu = 0$. The natural covariant derivative $\overline{D}$ on $\overline{\mathcal{M}}$ is simply [33,34]:

$$\overline{D}_\alpha = \overline{h}^\mu_\alpha \nabla_\mu. \tag{3}$$

---

[3]This was not shown in [30] but can easily be seen to be the case for the metrics discussed in [30] for which only a few multipoles are found to be well-defined. The reason the ACMC expansion fails in these cases is essentially because the energy-momentum tensor has an angular dependence that is "too strong" for its $1/r$ fall-off — e.g. $T_{\theta\phi} \sim \cos^3 \theta \sin^3 \theta / r^5$ (eq. (4.4) in [30]).

Note that the correct way to project a tensor covariant derivative is $\overline{D}_\alpha T_{\beta\gamma} = \overline{h}^\mu_\alpha \overline{h}^\nu_\beta \overline{h}^\rho_\gamma \nabla_\mu T_{\nu\rho}$.

We can also introduce a rescaled version of the three-dimensional metric, $h_{ab}$:

$$h_{ab} = -\xi^2 g_{ab} + \xi_a \xi_b\,. \tag{4}$$

When we consider the metric $h_{ab}$, we will denote the corresponding three-dimensional manifold $\mathcal{M}$, so that there is no possible confusion with $\overline{\mathcal{M}}$ which is endowed with the metric $\overline{h}$. The metric $h_{ab}$ can also be understood as the three-dimensional metric coming from a (timelike) Kaluza-Klein reduction over $\xi$. This amounts to introducing coordinates $(t, x^i)$ such that locally $\xi = \partial_t$ and the metric takes the standard Kaluza-Klein form:

$$g_{\mu\nu} dx^\mu dx^\nu = \lambda(dt + \mathcal{A})^2 - \lambda^{-1} h_{ij} dx^i dx^j\,. \tag{5}$$

In such coordinates, we have:

$$h_{ij} = -g_{00} g_{ij} + g_{0i} g_{0j}\,. \tag{6}$$

We will always denote $\mu, \nu, \cdots$ for four-dimensional indices; $a, b, \cdots$ for the induced three-dimensional indices on $\mathcal{M}$; $\alpha, \beta, \cdots$ for the three-dimensional indices on $\overline{\mathcal{M}}$, and finally $i, j, \cdots$ as well as $a_1, a_2, \cdots$ for the three-dimensional indices when we are using coordinates where $\xi = \partial_t$. Four-dimensional covariant derivatives will always be denoted with $\nabla_\mu$ and three-dimensional ones by $D$ (or $\overline{D}, \tilde{D}$). Unfortunately, there are many kinds of indices to keep track of, but fortunately it is usually clear from the context in what space (and with what metric) we are working in.

We can introduce the twist vector $\omega_\mu$, defined by:

$$\omega_\mu = \epsilon_{\mu\nu\rho\sigma} \xi^\nu \nabla^\rho \xi^\sigma\,. \tag{7}$$

Equivalently in form notation, $\boldsymbol{\omega} = *(\boldsymbol{\xi} \wedge d\boldsymbol{\xi})$. The curl of this vector is given by:

$$\partial_{[\mu} \omega_{\nu]} = -\epsilon_{\mu\nu\rho\sigma} \xi^\rho R^\sigma{}_\lambda \xi^\lambda\,. \tag{8}$$

Since the spacetime is a solution to the vacuum Einstein equations, $R_{\mu\nu} = 0$, this curl vanishes. It follows that the twist vector is derivable from a potential $\omega$:

$$\nabla_\mu \omega = \omega_\mu\,, \tag{9}$$

which defines the scalar field $\omega$. From the Kaluza-Klein point of view (5), the Kaluza-Klein vector $\mathcal{A}$ with field strength $\mathcal{F} = d\mathcal{A}$ is related to the twist vector $\omega_\mu$ in (7) as $\omega_t = 0$ and:

$$\omega_i = -\lambda^2 (*_3 \mathcal{F})_i\,, \tag{10}$$

where $*_3$ is the Hodge dual with respect to $h_{ij}$. The vanishing of the curl of $\omega_\mu$ is then simply the statement that the three-dimensional equation of motion for the gauge field $\mathcal{F}$ is sourceless, $d(*_3 \lambda^2 \mathcal{F}) = 0$, and the scalar $\omega$ is the three-dimensional dual gauge potential, $-d\omega = \tilde{\mathcal{F}} = \lambda^2 *_3 \mathcal{F}$. The Bianchi identity for $\mathcal{F}$ translates to an "equation of motion" for $\omega_i$:

$$D_i(\lambda^{-2} \omega^i) = 0\,. \tag{11}$$

The manifold $\mathcal{M}$ can be conformally transformed into a new three-manifold $\tilde{\mathcal{M}}$ with metric $\tilde{h}_{ab}$ by:

$$\tilde{h}_{ab} = \Omega^2 h_{ab}\,, \tag{12}$$

where $\Omega \sim 1/r^2$ at spatial infinity (with $r$ the distance from the object). In this way, spatial infinity of $\mathcal{M}$ is brought to a single point $\Lambda$, which is a regular point on the compact manifold $\tilde{\mathcal{M}}$.

The conformal factor $\Omega$, together with (local) coordinates at $\Lambda$ on $\tilde{\mathcal{M}}$, must be chosen such that the metric coefficients $\tilde{h}_{ab}$ and $\Omega$ are smooth (infinitely differentiable) functions of the coordinates at $\Lambda$. For stationary, vacuum spacetimes it is known that this is always possible, and that moreover one can choose coordinates such that $\tilde{h}_{ab}$ and $\Omega$ are analytic at $\Lambda$ [12–14, 35, 36]. (The demand of smoothness of $\tilde{h}_{ab}$ and $\Omega$ at $\Lambda$ may seem like a trivial demand, but it is not — see further in Section 3.4.)

Indeed, Geroch and Hansen used the existence of such a conformal factor $\Omega$ to provide a definition of asymptotic flatness. The manifold $\mathcal{M}$ with metric $h_{ab}$ (and thus the full stationary four-dimensional spacetime $g_{ab}$) is *asymptotically flat* if there exists a manifold $\tilde{\mathcal{M}}$ with metric $\tilde{h}_{ab}$ such that $\tilde{\mathcal{M}} = \mathcal{M} \cup \Lambda$ where $\Lambda$ is a single point; $\tilde{h}_{ab} = \Omega^2 h_{ab}$ is a smooth metric on $\tilde{\mathcal{M}}$; and $\Omega|_\Lambda = \tilde{D}_a \Omega|_\Lambda = 0$, $\tilde{D}_a \tilde{D}_b \Omega|_\Lambda = 2\tilde{h}_{ab}|_\Lambda$ [3]. We will also always assume this particular definition of asymptotic flatness.

We can now combine the scalar fields $\lambda, \omega$ into new scalar fields $\Phi_M, \Phi_J$ on $\mathcal{M}$:

$$\Phi_M = \frac{1}{4\lambda}(\lambda^2 + \omega^2 - 1), \qquad \Phi_J = \frac{1}{2\lambda}\omega, \tag{13}$$

of which we can introduce their conformal transformed version:

$$\tilde{\Phi}_{M,J} = \Omega^{-1/2}\Phi_{M,J}. \tag{14}$$

These scalar fields are then used to define symmetric and trace-free (STF) tensors $P^{M,J}_{a_1\cdots a_\ell}$ on $\tilde{\mathcal{M}}$ for every degree $\ell$ by $P^{M,J} = \tilde{\Phi}_{M,J}$ for $\ell = 0$, and then the recursive definition:

$$P^{M,J}_{a_1\cdots a_{\ell+1}} = \left[\tilde{D}_{a_{\ell+1}}P^{M,J}_{a_1\cdots a_\ell} - \frac{1}{2}\ell(2\ell-1)\tilde{R}_{a_\ell a_{\ell+1}}P^{M,J}_{a_1\cdots a_{\ell-1}}\right]^{\text{STF}}, \tag{15}$$

where the STF superscript means to take the symmetric, trace-free part, and $\tilde{D}, \tilde{R}$ are quantities defined on $\tilde{\mathcal{M}}$. The gravitational multipoles (which are themselves constant STF tensors) are then (finally) given by evaluating these tensors at the point $\Lambda$:[4]

$$M_{a_1\cdots a_\ell} = \frac{1}{(2\ell-1)!!}P^M_{a_1\cdots a_l}(\Lambda), \qquad S_{a_1\cdots a_\ell} = \frac{\ell+1}{2\ell(2\ell-1)!!}P^J_{a_1\cdots a_\ell}(\Lambda). \tag{16}$$

Note that the presence of the term proportional to the Ricci tensor $\tilde{R}_{ab}$ in (15) is not arbitrary. As Geroch showed, multipole moments transform among themselves in a specific way under a change of origin [1]. In the conformally compactified spacetime $\tilde{\mathcal{M}}$, such a change of origin corresponds to a change of conformal factor, $\Omega \rightarrow \Omega'$. Geroch [2] showed that (15) and (16) leads precisely to the correct transformation rules for the multipole moments only when the Ricci tensor term is included.

Finally, the multipoles defined in (16) are clearly only well defined if the conformal scalars $\tilde{\Phi}_{M,J}$ are smooth at $\Lambda$. As Hansen showed [3], their smoothness indeed follows directly from the Einstein equations, suitably rewritten. In particular, the four-dimensional vacuum Einstein equations are implied by following equations of motion for the scalars $\Phi_M, \Phi_J$ and the metric $h_{ab}$ on $\mathcal{M}$:

$$D^2\Phi_{M,J} - \frac{\mathcal{R}}{8}\Phi_{M,J} = \frac{15}{8}\kappa^4\Phi_{M,J},$$
$$\mathcal{R}_{ab} = 2\left[(D_a\Phi_M)(D_b\Phi_M) + (D_a\Phi_J)(D_b\Phi_J) - (D_a\psi)(D_b\psi)\right], \tag{17}$$

---

[4]I always use the Thorne normalization for the multipoles; the actual normalization used by Geroch-Hansen differs by precisely the prefactors in (16).

where $D_a, \mathcal{R}_{ab}$ are quantities defined with $h_{ab}$ on $\mathcal{M}$, and the scalars $\kappa, \psi$ are given by:

$$\kappa^4 = \frac{1}{2\lambda^2}\left[D_a\lambda D^a\lambda + D_a\omega D^a\omega\right], \tag{18}$$

$$\psi = \frac{1}{4\lambda}(\lambda^2 + \omega^2 + 1). \tag{19}$$

The equations (17) are conformally invariant elliptic equations if $\Phi_{M,J}$ and $\kappa$ have conformal dimension $-1/2$ and $\psi$ has dimension 0, i.e. that $\Phi_{M,J}$ transform precisely as (14). The conformal invariance of the elliptic equations (17) implies that $\tilde{\Phi}_{M,J}$ will be smooth everywhere on $\tilde{\mathcal{M}}$ — and in particular at $\Lambda$ — if and only if the conformally transformed coefficient $\tilde{\kappa} = \Omega^{-1/2}\kappa$ is smooth. From (18), the elliptic conformally invariant equation of motion for $\kappa$ can also be derived, leading eventually to a proof of smoothness of all quantities $\tilde{\kappa}, \tilde{\Phi}_{M,J}$ involved in the transformed version of (17) at $\Lambda$, ensuring that the multipoles defined in Section 2 are well-defined [3].

## 2.2 Thorne ACMC-coordinates and multipoles

While elegant and manifestly coordinate invariant, the Geroch-Hansen formalism of extracting multipoles can be rather cumbersome in practice; for example, it requires a judicious choice of the conformal factor $\Omega$. Perhaps less elegant but much more practical is Thorne's formalism [4]. Thorne introduces "asymptotically Cartesian and mass-centered" (ACMC) coordinates for asymptotically flat, stationary, and vacuum spacetimes. In such a coordinate system $(t, x^i)$, the metric has the following asymptotic expansion:

$$g_{00} = -1 + \frac{2M}{r} + \frac{\mathcal{S}_0}{r^2} + \sum_{\ell=2}^{\infty}\frac{1}{r^{\ell+1}}\left[\frac{2(2\ell-1)!!}{\ell!}M_{A_\ell}N_{A_\ell} + \mathcal{S}_{\ell-1}\right],$$

$$g_{0j} = \sum_{\ell=1}^{\infty}\frac{1}{r^{\ell+1}}\left[-\frac{4\ell(2\ell-1)!!}{(\ell+1)!}\epsilon_{jka_\ell}S_{kA_{\ell-1}}N_{A_\ell} + \mathcal{S}_{\ell-1}\right], \tag{20}$$

$$g_{ij} = \delta_{ij} + \sum_{\ell=0}^{\infty}\frac{\mathcal{S}_\ell}{r^{\ell+1}},$$

where $r = \sqrt{(x^1)^2 + (x^2)^2 + (x^3)^2}$ is the usual radius, and we use the shorthands $A_\ell = a_1\cdots a_\ell$ and $N_{A_\ell} = n_{a_1}\cdots n_{a_\ell}$, where $n_i = x^i/r$. Moreover, $\epsilon_{ijk}$ is the three-dimensional (flat) Levi-Civita symbol, and we always sum over repeated $i, j, \cdots$ indices. The symbol $\mathcal{S}_\ell$ denotes an arbitrary angular dependence on spherical harmonics of maximal order $\ell$. In other words, $\mathcal{S}_\ell$ can contain dependence on $n_i$ vectors through products such as $N_{A_\ell}, N_{A_{\ell-1}}, \cdots$ but cannot depend on $N_{A_{\ell+1}}$ or any other higher-order angular dependence, i.e.:

$$\mathcal{S}_\ell = \sum_{\ell'\leq\ell}c_{A_{\ell'}}N_{A_{\ell'}}, \tag{21}$$

for arbitrary constant tensors $c_{A_\ell}$. Note that the symbol $\mathcal{S}$ is used as a placeholder; the $\mathcal{S}_\ell$ quantities appearing in each component of the metric can be different.

The first crucial condition that an acceptable ACMC coordinate system satisfies is the angular dependence at every order in the asymptotic expansion. For example, for $g_{00}$, the term at order $r^{-(\ell+1)}$ can have angular dependence on spherical harmonics up to the maximal order $\ell$, but not higher. A coordinate system which has a higher order angular dependence than "allowed" is not ACMC.[5] The second condition for an ACMC coordinate system — the "mass

---

[5]Thorne actually discusses a generalization of this condition by defining ACMC-$N$ coordinates, where $N$ is roughly the first order $r^{-(N+1)}$ in the asymptotic expansion where the ACMC condition fails. In ACMC-$N$ coordinates, the multipoles up to order $N+1$ can still be read off from the metric expansion.

centered" part — is that there is no mass dipole tensor, $M_{a_1} = 0$, and equivalently the $r^{-2}$ term in $g_{00}$ must be a constant.[6]

The simplicity of the ACMC coordinate expansion is that the multipole tensors $M_{A_\ell}, S_{A_\ell}$ can simply be read off from the $r^{-(\ell+1)}$ terms in the expansions of $g_{00}$ and $g_{0j}$. Different ACMC coordinate systems may differ in the lower-order angular dependences $\mathcal{S}_\ell$, but will always agree on the values of the multipoles, as shown by Thorne [4].

## 2.3 Equivalence of Geroch-Hansen and Thorne formalisms

The equivalence of the Geroch-Hansen and Thorne formalisms was shown by Gürsel [5]. The proof starts by assuming a coordinate system in which the metric is ACMC as in (20) and also harmonic, $\partial_\mu(\sqrt{-g}\, g^{\mu\nu}) = 0$. (For vacuum spacetimes, it is always possible to find such a "canonical harmonic gauge" that is ACMC, see also the discussion in Section 3.4.) Then, using these coordinates, the relevant Geroch-Hansen quantities are calculated on the induced three-dimensional manifold $\mathcal{M}$. This includes the induced metric $h_{ij}$ of (4) and its inverse:

$$h_{ij} = \delta_{ij} + \sum_{\ell=2}^\infty \frac{\mathcal{S}_\ell}{r^{\ell+1}}, \qquad h^{ij} = \delta^{ij} + \sum_{\ell=2}^\infty \frac{\mathcal{S}_\ell}{r^{\ell+1}}, \tag{22}$$

which Gürsel shows is also automatically harmonic, $\partial_i(\sqrt{h}\, h^{ij}) = 0$. The scalar field $\lambda$ is simply $g_{00}$ of (20). The twist form $\omega_\mu$ is given by $\omega_0 = 0$ and:

$$\omega_i = -\sum_{\ell=1}^\infty \left( \frac{4\ell(2\ell-1)!!}{(\ell+1)!} \epsilon_{ijk}\epsilon_{jma_\ell} S_{mA_{\ell-1}} \partial_k \left[ \frac{N_{A_\ell}}{r^{\ell+1}} \right] + \frac{\mathcal{S}_{\ell-1}}{r^{\ell+1}} \right) \tag{23}$$

(In (22), (23), and (25), I correct typos in eqs. (30)-(31), (34), and (49) of [5]). Integrating this gives the scalar field potential $\omega$ (through $\nabla_\mu \omega = \omega_\mu$):

$$\omega = -\sum_{\ell=1}^\infty \frac{1}{r^{\ell+1}} \left[ \frac{4\ell(2\ell-1)!!}{(\ell+1)!} S_{A_\ell} N_{A_\ell} + \mathcal{S}_{\ell-1} \right]. \tag{24}$$

To show that (24) follows from integrating (23), the identities in Appendix A are useful.

The scalar fields $\Phi_{M,J}$ can now easily be computed explicitly from $\lambda$ in (20) and $\omega$ in (24). Gürsel then chooses a conformal factor and altered coordinates $\tilde{x}^i$ on the compactified manifold $\tilde{\mathcal{M}}$ in such a way that:

$$\tilde{x}^i = \frac{x^i}{r^2} + \frac{1}{r}\sum_{\ell\le2}\frac{\mathcal{S}_{\ell-1}}{r^\ell}, \qquad \Omega = \frac{1}{r^2}\left[1 + \sum_{\ell\le2}\frac{\mathcal{S}_{\ell-1}}{r^\ell}\right], \qquad \tilde{h}_{\tilde{i}\tilde{j}} = \delta_{\tilde{i}\tilde{j}} + \sum_{\ell\le2}\tilde{r}^\ell \mathcal{S}_{\ell-1}. \tag{25}$$

It is critical to be able to choose the coordinates $\tilde{x}^i$ and $\Omega$ such that $\Omega, \tilde{h}_{\tilde{i}\tilde{j}}, \tilde{\Phi}_{M,J}$ are analytic functions of $\tilde{x}^i$ and that (25) is satisfied. Crucially, this relies (besides various mathematical lemmas that Gürsel proves) on a property of the Beig-Simon conformal factor [5, 12–14]:

$$\Omega^{(\mathrm{BS})} = \frac{1}{2}B^2\left[\left(1 + 4\Phi_M^2 + 4\Phi_J^2\right)^{1/2} - 1\right], \tag{26}$$

where $B$ is a constant chosen such that $(\tilde{D}_i\tilde{D}_j\Omega^{(\mathrm{BS})})(\Lambda) = 2\tilde{h}_{ij}^{(\mathrm{BS})}(\Lambda)$. Specifically, for a choice of coordinates $\tilde{x}^i$ such that $\tilde{h}_{\tilde{i}\tilde{j}}^{(\mathrm{BS})}$ is harmonic, all of $\Omega^{(\mathrm{BS})}, \tilde{h}_{\tilde{i}\tilde{j}}^{(\mathrm{BS})}, \tilde{\Phi}_{M,J}^{(\mathrm{BS})}$ are analytic functions of

---

[6]It is easy to relax this mass-centered condition to have AC coordinates where the dipole is not necessarily zero. The resulting multipoles $\tilde{M}_{A_\ell}, \tilde{S}_{A_\ell}$ that are read off from a metric in AC coordinates are easily related to the true multipoles $M_{A_\ell}, S_{A_\ell}$ [17].

$\tilde{x}^i$. This is shown by Gürsel (in the Appendix of [5]) by explicitly calculating the subleading angular dependences in the ACMC expansion using the vacuum Einstein equations.

Finally, using (25), Gürsel computes the tensors $P_{A_l}^{M,J}$ and the multipoles $M_{A_\ell}, S_{A_\ell}$. The result, of course, is that the multipoles calculated in (16) precisely agree with those in the ACMC expansion (20) [5].

# 3 Generalization to arbitrary spacetimes

This section explains how the Geroch-Hansen formalism as well as the proof of its equivalence with the Thorne formalism can be generalized to arbitrary *non-vacuum* stationary spacetimes.

We will assume that the metric is governed by Einstein's equations $R_{\mu\nu} - (1/2)Rg_{\mu\nu} = T_{\mu\nu}$. Note that this includes solutions in higher-derivative theories of gravity, as long as the solution considered is perturbatively linear in the higher-derivative coupling [22]. We demand that the matter content is such that the metric is asymptotically flat. Note that we are still using the definition of asymptotic flatness using the existence of the conformally compactified three-dimensional spacetime as introduced by Geroch and discussed in Section 2.1, even though the spacetime is no longer vacuum. Of course, this condition of asymptotic flatness implicitly demands specific asymptotic fall-offs for the energy momentum tensor through the Einstein equations (see e.g. (69)).

We will assume that the metric (both the four-dimensional metric $g_{\mu\nu}$ and the three-dimensional compactified metric $\tilde{h}_{ij}$) are sufficiently smooth in the chosen (harmonic) coordinates in order for the Geroch-Hansen formalism to apply; we will also assume the existence of a (harmonic) ACMC coordinate system with an expansion of the form (20), especially in Section 3.3.2. These assumptions of smoothness and the existence of ACMC coordinate systems intuitively seem rather mild, but are crucial for the application of the Geroch-Hansen formalism; this is further discussed in Section 3.4.

We will first detail the general definition and construction of the improved twist vector $\omega_\mu^I$ that allows for the definition of the twist scalar $\omega$ in non-vacuum spacetimes in Section 3.1. Section 3.2 discusses the precise conditions necessary for the *existence* of $\omega_\mu^I$. Section 3.3 then deals with the non-uniqueness of the improved twist vector $\omega_\mu^I$ (as defined in Section 3.1) and describes the necessary "gauge-fixing" to fix $\omega_\mu^I$ (and thus $\omega$) so that the multipoles are well-defined. Finally, Section 3.4 discusses the conditions for the existence of an ACMC coordinate system.

## 3.1 Construction of the improved twist vector

An immediate problem with generalizing the Geroch-Hansen formalism to non-vacuum spacetimes is the definition of the twist vector potential $\omega$. Defining $\omega_\mu$ through (7) means its curl (8) is proportional to the Ricci tensor:

$$\omega_\mu = \epsilon_{\mu\nu\rho\sigma}\xi^\nu\nabla^\rho\xi^\sigma, \qquad \partial_{[\mu}\omega_{\nu]} = -\epsilon_{\mu\nu\rho\sigma}\xi^\rho R^\sigma{}_\lambda \xi^\lambda. \tag{27}$$

For non-vacuum spacetimes with $R_{\mu\nu} \neq 0$, this curl then does not vanish, and it is not possible to define the scalar field potential $\omega$ as $\nabla_\mu \omega = \omega_\mu$.

The solution to this conundrum lies with a definition of an "improvement" vector $\omega_\mu^I$, such that the "improved" total twist vector has vanishing curl, $\partial_{[\mu}\omega_{\nu]}^{(\text{tot})} = 0$ with $\omega_\mu^{(\text{tot})} = \omega_\mu + \omega_\mu^I$; then we can define the twist scalar $\omega$ through

$$\nabla_\mu \omega = \omega_\mu^{(\text{tot})}, \tag{28}$$

and apply the Geroch-Hansen formalism again with the scalars $\lambda$ and $\omega$. I will discuss the construction of this improvement vector for Einstein-Maxwell theory (which is known) and its immediate extension to more general $\mathcal{N} = 2$ STU supergravity theories (which was not yet known) before discussing the fully general case for arbitrary matter.

### 3.1.1 Einstein-Maxwell and STU supergravities

The Ricci tensor does not vanish in solutions to Einstein-Maxwell theory, with Lagrangian density:

$$\mathcal{L} = R - \frac{1}{4} F_{\mu\nu} F^{\mu\nu}. \tag{29}$$

However, it was shown [7] that for stationary solutions where the Maxwell field is also stationary, $\mathcal{L}_\xi F = 0$ (so that we can also take $\mathcal{L}_\xi A = 0$ — see Appendix B.1), one can define an "improvement vector" $\omega_\mu^I$ from the electromagnetic field, for which the curl of $\omega_\mu^{(\text{tot})} = \omega_\mu + \omega_\mu^I$ vanishes and the scalar $\omega$ can be defined as $\nabla_\mu \omega = \omega_\mu^{(\text{tot})}$. It is convenient to first introduce the dual field strength $\tilde{F}$ in the usual way, $\tilde{F} = *F$, so:

$$\tilde{F}_{\mu\nu} = \frac{1}{2} \epsilon_{\mu\nu\rho\sigma} F^{\rho\sigma}. \tag{30}$$

The Bianchi identity is $dF = 0$, and assures us that $F$ can (locally) be written in terms of a potential as $F = dA$. The equations of motion for the Maxwell field can be written as $d\tilde{F} = 0$, so the dual field $\tilde{F}$ can also be written in terms of a dual potential, $\tilde{F} = d\tilde{A}$. Define the component along $\xi$ of both potentials as $\rho, \tilde{\rho}$, so:

$$\rho \equiv \xi \cdot A, \qquad \tilde{\rho} \equiv \xi \cdot \tilde{A}. \tag{31}$$

Of course, in coordinates where $\xi = \partial_t$, these potentials $\rho, \tilde{\rho}$ correspond with the electrostatic potentials $A_t, \tilde{A}_t$. Note that $\rho$ (resp. $\tilde{\rho}$) do *not* change under gauge transformations that preserve $\mathcal{L}_\xi A = 0$ (resp. $\mathcal{L}_\xi \tilde{A} = 0$), so they are unambiguously well-defined quantities; see Appendix B.2. The improvement vector is then given by [7]:

$$\omega_\mu^I = -\frac{1}{2} \left( \tilde{\rho} \nabla_\mu \rho - \rho \nabla_\mu \tilde{\rho} \right), \tag{32}$$

which in form notation is:

$$\boldsymbol{\omega}^I = -\frac{1}{2} \left( \tilde{\rho} \, d\rho - \rho \, d\tilde{\rho} \right). \tag{33}$$

Note that:

$$d\boldsymbol{\omega}^I = -d\tilde{\rho} \wedge d\rho, \tag{34}$$

which can be shown to be the necessary improvement vector such that $\omega_\mu^{(\text{tot})} = \omega_\mu + \omega_\mu^I$ satisfies $d\boldsymbol{\omega}^{(\text{tot})} = 0$ [7].

The construction of the improved twist vector can be readily generalized to four-dimensional $\mathcal{N} = 2$ supergravity including an arbitrary number of vector multiplets; this includes the STU supergravity theories — the arena where gravitational multipoles of many black holes and horizonless, smooth microstate geometries have been considered [15–19]. We start with the (bosonic) Lagrangian density:

$$\mathcal{L} = R - 2g_{IJ} \partial_\mu z^I \partial^\mu \bar{z}^J - \frac{1}{4} I_{\Lambda\Sigma} F_{\mu\nu}^\Lambda F^{\Sigma,\mu\nu} + \frac{1}{4} R_{\Lambda\Sigma} F_{\mu\nu}^\Lambda \tilde{F}^{\Sigma,\mu\nu}, \tag{35}$$

where $\tilde{F} = *F$ is the Hodge dual of $F$. There are $n + 1$ gauge fields $F^\Lambda = dA^\Lambda$, $\Lambda \in \{0, 1, 2, \cdots, n\}$ and $n$ complex scalar fields $z^I$, $I = 1, 2, \cdots, n$. The matrices $I, R$ are symmetric, real $(n + 1) \times (n + 1)$ matrices which depend on the scalars — we will not need to be

concerned with their precise form (but see e.g. appendix A in [19]). Note that the scalars are not charged under the gauge fields. We assume that both the gauge fields and the scalars are stationary, so $\mathcal{L}_\xi F = \mathcal{L}_\xi z^I = 0$. The improvement twist form $\omega_\mu^I$ is then given by:

$$\omega_\mu^I \equiv -\frac{1}{2}(\tilde{\rho}_\Lambda \nabla_\mu \rho^\Lambda - \rho^\Lambda \nabla_\mu \tilde{\rho}_\Lambda), \tag{36}$$

or $\boldsymbol{\omega}^I = -(\tilde{\rho}_\Lambda d\rho^\Lambda - \rho^\Lambda d\tilde{\rho}_\Lambda)/2$, where $\rho^\Lambda = \xi^\mu A_\mu^\Lambda$ is the component of the gauge fields along $\xi$, and similarly $\tilde{\rho}_\Lambda = \xi^\mu \tilde{A}_\Lambda$ is the component of the dual gauge field along $\xi$; note that these potentials $\rho^\Lambda, \tilde{\rho}_\Lambda$ are well-defined and not ambiguous under gauge transformations, analogous to the Einstein-Maxwell case. The precise definition of the dual gauge fields and more details on the derivation of (36) is given in Appendix B. As far as I know, this is the first presentation of the Geroch-Hansen improvement vector (36) for $\mathcal{N} = 2$ supergravity theories.

### 3.1.2 General matter

For general matter content, the status of the Geroch-Hansen multipole formalism was not clear up until now. For example, if there could exist matter couplings for which no improvement vector could be found, then this may even have implied the existence of a *third* family of multipoles (besides the "usual" two $M_{A_\ell}, S_{A_\ell}$) [11]. Luckily, as I show here, it is relatively simple to show that a stationary solution in a theory with any matter content allows for the construction of an improvement vector $\omega_\mu^I$, such that $\omega_\mu^{(\text{tot})} = \omega_\mu + \omega_\mu^I$ is curl-less and allows for the definition of the twist scalar $\omega$.

We start by defining the vector $V^\mu$ using the energy-momentum tensor $T^{\mu\nu}$ and the (projection) spatial metric (2) on $\overline{\mathcal{M}}$:

$$V^\mu = \overline{h}_\sigma^\mu T^{\sigma\nu}\xi_\nu = T^{\mu\nu}\xi_\nu - \lambda^{-1}\xi^\mu\xi_\nu T^{\nu\rho}\xi_\rho. \tag{37}$$

This vector is divergenceless, $\nabla_\mu V^\mu = 0$, since $\xi$ is Killing and energy-momentum is conserved ($\nabla_\mu T^{\mu\nu} = 0$). Note that the energy-momentum tensor must also be stationary, $\mathcal{L}_\xi T^{\mu\nu} = 0$, since the energy-momentum tensor is related through the Einstein equations to derivatives of the stationary metric. The Hodge dual of the one-form $V$ is closed, $d * V = 0$. Since also $\mathcal{L}_\xi(*V) = 0$ and using the form identity $\mathcal{L}_\xi = di_\xi + i_\xi d$, we find that:

$$dW^{(2)} = 0, \qquad W^{(2)} \equiv i_\xi * V. \tag{38}$$

So $W^{(2)}$ is a closed two-form, which moreover lives on $\overline{\mathcal{M}}$ (or $\mathcal{M}$) — this can be seen from $i_\xi W^{(2)} = 0$ so that $W_{\mu\nu} = \overline{h}_\mu^\alpha W_{\alpha\nu}$.

We have seen that the form $W^{(2)}$ is certainly a *closed* two-form on the three-dimensional manifold $\overline{\mathcal{M}}$; under mild and reasonable assumptions (see below!), it will also be an *exact* form as well, meaning a one-form $B^{(1)}$ exists (on $\overline{\mathcal{M}}$) such that:

$$W^{(2)} = \overline{d}B^{(1)}, \tag{39}$$

where we denote $\overline{d}$ to denote the exterior derivative on $\overline{\mathcal{M}}$; or in components:

$$\left(i_\xi * V\right)_{\mu\nu} = \epsilon_{\mu\nu\rho\sigma}\xi^\rho V^\sigma = W_{\mu\nu}^{(2)} = 2\partial_{[\mu}B_{\nu]}^{(1)}. \tag{40}$$

Note that we do not need to distinguish between indices on $\overline{\mathcal{M}}$ or on the full four-dimensional spacetime, as (40) is valid on either manifold. It follows almost immediately that our sought-after improvement vector $\omega_\mu^I$ is simply:

$$\omega_\mu^I = 2B_\mu^{(1)}. \tag{41}$$

The curl of (41) is:

$$\partial_{[\mu}\omega^I_{\nu]} = 2\partial_{[\mu}B^{(1)}_{\nu]} = \epsilon_{\mu\nu\rho\sigma}\xi^\rho V^\sigma = \epsilon_{\mu\nu\rho\sigma}\xi^\rho T^\sigma{}_\lambda \xi^\lambda. \tag{42}$$

From (27), (42), and the Einstein equations $R_{\mu\nu} - Rg_{\mu\nu}/2 = T_{\mu\nu}$, it follows that $\omega^{(\text{tot})}_\mu = \omega_\mu + \omega^I_\mu$ has vanishing curl and allows for the definition of the twist scalar $\omega$ through $\nabla_\mu\omega = \omega^{(\text{tot})}_\mu$.

**Equations of motion for $\Phi_{M,J}$**    It is interesting to report the alteration of the equations of motion (17) in the presence of matter, which can be written as follows:

$$\begin{aligned}
D^2\Phi_M - \frac{\mathcal{R}}{8}\Phi_M &= \frac{15}{8}\kappa^4\Phi_M + \kappa'\Phi_M + \sigma\,, \\
D^2\Phi_J - \frac{\mathcal{R}}{8}\Phi_J &= \frac{15}{8}\kappa^4\Phi_J + \kappa'\Phi_J\,, \\
\mathcal{R}_{ab} &= 2\left[(D_a\Phi_M)(D_b\Phi_M) + (D_a\Phi_J)(D_b\Phi_J) - (D_a\psi)(D_b\psi)\right] \\
&\quad + (T_{ab} - h_{ab}T^c{}_c) + \frac{1}{2\lambda^2}\left(\omega^I_a\omega^I_b - 2\omega^I_{(a}D_{b)}\omega\right),
\end{aligned} \tag{43}$$

where $T_{ab} = \bar{h}^\mu_a T_{\mu\nu}\bar{h}^\nu_b$ is the energy-momentum tensor projected onto three dimensions. The new quantities $\kappa', \sigma$ are also determined by the energy-momentum tensor. We can first define the following shorthands:

$$\mathcal{T}_1 = \frac{T_{tt}}{\lambda^2}, \qquad \mathcal{T}_2 = T^a{}_a\,, \qquad \mathcal{T}_3 = \frac{1}{\lambda^2}\omega^I_a(\omega^{I,a} - 2D^a\omega)\,, \tag{44}$$

where $T_{tt}$ is calculated in coordinates where $\xi = \partial_t$. Then $\kappa', \sigma$ are given by:

$$\kappa' = \alpha_1\mathcal{T}_1 + \alpha_2\mathcal{T}_2 + \frac{15}{16}\mathcal{T}_3\,, \tag{45}$$

$$\sigma = \alpha_3\mathcal{T}_1 + \alpha_4\mathcal{T}_2 - \frac{\lambda}{2}\mathcal{T}_3\,. \tag{46}$$

The Einstein equations only fix that:[7]

$$-\lambda^2 + \alpha_1\left(\lambda^2 - \omega^2 - 1\right) + 4\lambda\alpha_3 - \omega^2 - 1 = -5\lambda^2 + 4\alpha_2\left(\lambda^2 - \omega^2 - 1\right) + 16\lambda\alpha_4 - 3\omega^2 - 3 = 0, \tag{47}$$

but otherwise leave undetermined the coefficients $\alpha_{1-4}$, which are a priori functions of $\lambda$ and $\omega$ (although note that solutions to (47) exist where the $\alpha_i$ are pure constants).

The linearized limit offers more insight into the unknowns $\alpha_i$. In this limit, assuming $\alpha_i$ remain finite, we must have:

$$(\alpha_3)_{\text{lin}} = (\alpha_4)_{\text{lin}} = -\frac{1}{2}\,, \tag{48}$$

and then we also immediately retrieve the linearized version of (43):

$$D^2\Phi_M = -\frac{1}{2}\left(T_{tt} + T^c{}_c\right), \qquad D^2\Phi_J = 0, \qquad \mathcal{R}_{ab} = T_{ab} - h_{ab}T^c{}_c\,. \tag{49}$$

So, in the linearized theory, $\Phi_J$ reduces to a source-less Newtonian potential, while $\Phi_M$ reduces to a Newtonian potential with source $-(T_{tt} + T^c{}_c)/2$. In the slow-moving limit $|T_{ab}| \ll |T_{tt}|$,

---

[7]The four-dimensional Einstein equations decompose into the $(tt),(ta),(ab)$ components. The $(ta)$ components are solved by the introduction of $\omega^I_i$; the $(ab)$ components are solved (unambiguously) by $\mathcal{R}'_{ab}$. The equations of motion for $\Phi_{M,J}$ are only needed to solve the scalar $(tt)$ component of the Einstein equations, and so there is necessarily some ambiguity with how the terms in this one equation are "divided" among the two equations of motion for $\Phi_{M,J}$. Note that we did already fix the coefficient of the term $\sim \omega^I_a$ in (45)-(46) to a reasonable numerical coefficient (which was also explicitly verified for Einstein-Maxwell theory).

the equation of motion for $\Phi_M$ further reduces to the correct one for (half of) a Newtonian gravitational potential with matter distribution $T_{tt}$.

The "equation of motion" (11) must still be valid for the twist vector $\omega_a$. Using (43), we see that this implies the following equation of motion for $\boldsymbol{\omega}^I$:

$$D_a(\lambda^{-2}\omega^{I,a}) = 2\frac{\omega}{\lambda}\frac{(\lambda-2\alpha_3)\mathcal{T}_1 + (\lambda-2\alpha_4)\mathcal{T}_2}{1+\omega^2-\lambda^2}, \tag{50}$$

where we already used (47). Note that the linearized version of this equation is, using (48):

$$\partial_a\omega^{I,a} = 0. \tag{51}$$

The Einstein equations thus do not completely determine the unknowns $\alpha_i$ for an arbitrary unspecified energy-momentum tensor $T_{\mu\nu}$. Most likely, these will be further determined when a particular matter theory is specified; e.g. Einstein-Maxwell theory gives the following additional relation:

$$1 + 2\alpha_3\lambda + 2\alpha_4\lambda + \omega^2 - 3\lambda^2 = 0, \tag{52}$$

but there is no a priori reason the same relation would hold for other theories of matter.

Finally, the modified equations (43) suggest that $\kappa'$, resp. $\sigma$, should be a scalar with conformal dimension $-2$, resp. $-5/2$. The smoothness of the conformally transformed $\tilde{\Phi}_{M,J}$ (and thus the well-definedness of the multipole moments) then depends crucially on the smoothness of the conformally transformed energy-momentum coefficients $\tilde{\kappa}' = \Omega^{-2}\kappa', \tilde{\sigma} = \Omega^{-5/2}\sigma$.

There are two lingering concerns with the improved twist vector as introduced here. First of all, when is $\boldsymbol{B}^{(1)}$ in (39) well-defined, i.e. when is the *closed* two-form $\boldsymbol{W}^{(2)} = i_\xi * \boldsymbol{V}$ also *exact* on $\overline{\mathcal{M}}$? And secondly, $\boldsymbol{B}^{(1)}$ as defined by $\boldsymbol{W}^{(2)} = d\boldsymbol{B}^{(1)}$ is not unique as it can be shifted by an exact form, $\boldsymbol{B}^{(1)} \to \boldsymbol{B}^{(1)} + dA^{(0)}$, which leads to a shift in the twist scalar, $\omega \to \omega + 2A^{(0)}$, which could a priori shift the resulting (current) multipole tensors; what is the correct "gauge" choice for $\boldsymbol{B}^{(1)}$?[8]

I will address these two important issues in the following two subsections, including a discussion on the conditions that the energy-momentum must satisfy in order for $\boldsymbol{W}^{(2)}$ to be exact, and the correct "gauge-fixing" conditions for $\boldsymbol{B}^{(1)}$ that give rise to the correct twist scalar $\omega$. An immediate consequence of the analysis in Section 3.3.2 will be the equivalence of the Geroch-Hansen and Thorne formalisms, as long as an ACMC expansion exists and the prescribed "gauge-fixing" of $\boldsymbol{B}^{(1)}$ is adhered to.

## 3.2 Conditions on exactness of $\boldsymbol{W}^{(2)}$

When is the *closed* two-form $\boldsymbol{W}^{(2)} = i_\xi * \boldsymbol{V}$ also *exact* on $\overline{\mathcal{M}}$ (or equivalently on $\mathcal{M}$)? To gain some insight in this question, I first discuss how $\boldsymbol{W}^{(2)}$ and $\boldsymbol{B}^{(1)}$ arise in Einstein-Maxwell theory, before showing that $\boldsymbol{B}^{(1)}$ is always exact under certain mild assumptions on the topology of $\overline{\mathcal{M}}$.

### 3.2.1 Einstein-Maxwell (and $\mathcal{N} = 2$)

We can easily verify that $\boldsymbol{W}^{(2)}$ is exact in Einstein-Maxwell theory. Indeed, we found that (34), so:

$$d\boldsymbol{\omega}^I = -d\tilde{\rho} \wedge d\rho = 2\boldsymbol{W}^{(2)}. \tag{53}$$

Using that the energy-momentum tensor for Einstein-Maxwell can be expressed as:

$$T_{\mu\nu} = \frac{1}{4}\left(F_\mu{}^\rho F_{\nu\rho} + \tilde{F}_\mu{}^\rho \tilde{F}_{\nu\rho}\right), \tag{54}$$

---

[8]I would like to thank the anonymous referees for stressing that this point required addressing, which in particular led to the writing of Section 3.3.

and noting that $d\rho = i_\xi F$ (i.e. $(d\rho)_\mu = \xi^\nu F_{\nu\mu}$) and similarly $d\tilde{\rho} = i_\xi \tilde{F}$, it is relatively straightforward to calculate that indeed:

$$i_\xi * V = W^{(2)} = -\frac{1}{2} d\tilde{\rho} \wedge d\rho \,, \tag{55}$$

where $V$ is the vector defined through (37); we see that (55) is indeed consistent with (53) and $d\omega^I = 2W^{(2)}$, i.e. (41). For Einstein-Maxwell theory, then, the closed form $W^{(2)}$ can be written as the wedge product of two exact one-forms, and so is guaranteed to be exact itself. This means (39) always holds in Einstein-Maxwell theory, regardless of the particular solution that is considered. (The same reasoning naturally applies for STU supergravity.)

### 3.2.2 General matter

Even without knowledge of the matter content of the theory, we can still provide a precise prescription for when $W^{(2)}$ is exact. Since $W^{(2)}$ is defined on the three-dimensional Riemannian (non-compact) manifold $\overline{\mathcal{M}}$, (co)homology tells us that $W^{(2)}$ is exact if and only if the integral of $W^{(2)}$ over all non-trivial two-cycles of $\overline{\mathcal{M}}$ vanishes. (This follows from Poincaré duality, which holds as long as $\overline{\mathcal{M}}$ is a manifold with a *finite good cover*, which is a mild technical assumption that I will assume holds.[9]) Note that we are using results on the cohomology of non-compact *Riemannian* manifolds, which is the reason why it was important that $V$ in (37) and $W^{(2)}$ in (38) live on the three-dimensional manifold $\overline{\mathcal{M}}$.

Consider any spacetime where there is a *single* (up to homology) non-trivial two-cycle in $\overline{\mathcal{M}}$ — for example, a (single) stationary black hole with spherical horizon topology is such a geometry. The non-trivial two-cycle can be taken to be the two-sphere at spatial infinity. (This is homologous to the horizon two-sphere.) On this two-sphere, we see that:

$$\int_{S^2(\infty)} W^{(2)} \sim \int_{S^2(\infty)} \sqrt{-g}\, T_{0r}\, d\Omega_2 \sim \lim_{r\to\infty} \int (r^2 T_{0r}) \sin\theta\, d\theta d\phi = 0 \,, \tag{56}$$

where we introduced asymptotically spherical coordinates $(r, \theta, \phi)$. The last equality follows from asymptotic flatness.[10] Physically, this integral represents the total radial-pointing momentum flux through the sphere at infinity — which is clear should vanish in stationary spacetimes. We conclude that in any stationary spacetime with a *single* (up to homology) non-trivial two-cycle, $W^{(2)}$ will always be *exact* and (39) holds.

For a more general spacetime, (56) must still hold from asymptotic flatness; the integral of $W^{(2)}$ will vanish on the two-sphere at infinity. If there any additional non-equivalent two-cycles in the spacetime, there does not appear to be a generic argument ensuring that the integral of $W^{(2)}$ on it will always vanish. However, it does seem unlikely that spacetimes could exist containing two-cycles where the integral of $W^{(2)}$ over them does not vanish. In such spacetimes, a similar calculation to (56) implies that there would be a non-zero flux of radial (i.e. perpendicular to the two-cycle) momentum going through this two-cycle, which seems irreconcilable with the stationarity of the spacetime.

---

[9]A good cover is an open covering of $\overline{\mathcal{M}}$ where all non-empty finite intersections of opens in the cover are diffeomorphic to $\mathbb{R}^n$. Note that every manifold has a good cover, and a compact manifold always has a finite good cover. So, the quoted results in cohomology are a natural generalization to non-compact manifolds of the more familiar results of Poincaré duality for *compact* manifolds. See §5 in [37].

[10]The fall-off $T_{0r} \sim \mathcal{O}(r^{-3})$ is a straightforward consequence of the ACMC expansion (20), the Einstein equations, and (A.1). It can also be seen from the asymptotic Bondi expansion [38], where (using $u = t - r$) $T_{ur} \sim \mathcal{O}(r^{-4})$, and stationarity of the solution means that $T_{uu} \sim \mathcal{O}(r^{-3})$, so that $T_{tr} \sim \mathcal{O}(r^{-3})$. Without stationarity, $T_{uu} \sim T_{tr} \sim \mathcal{O}(r^{-2})$ would carry information of the Bondi news tensor, see (5.2.8) and (5.2.9) in [38].

### 3.3 The improvement form "gauge" choice

The two-form $W^{(2)} = i_\xi * V$ constructed above is unique and unambiguous, but the one-form potential $B^{(1)}$ is only defined through its exterior derivative giving $W^{(2)}$:

$$dB^{(1)} = W^{(2)}. \tag{57}$$

From the definition (57) alone, it appears that $B^{(1)}$ (and thus $\omega^I = 2B^{(1)}$ is not unique and can be shifted by a "gauge" transformation $B^{(1)} \to B^{(1)} + dA^{(0)}$ for a scalar $A^{(0)}$; this would result in the twist scalar being shifted as $\omega \to \omega + 2A^{(0)}$. Clearly, such a shift of twist scalar could result in the multipole structure being altered — for example, choosing $A^{(0)} = -\omega/2$ would result in all vanishing current multipoles.

There is thus a need to choose the "correct gauge" for $B^{(1)}$ in the construction of $\omega^I$. Since the gravitational multipoles should be a property of the metric alone, a natural choice is to choose $\omega^I$ such that the resulting multipoles only depend on the metric and not on the energy-momentum tensor (except, of course, indirectly as the metric solves the non-vacuum Einstein equations. In the following two subsections, we will explore how to make this notion precise. In the context of linearized gravity, we can find explicit coordinate-independent statements that fix the gauge of $\omega^I$ — in particular, (51) and the vanishing of all integrals (60); in the full non-linear theory we can use ACMC coordinates of the metric to fix the gauge of $\omega^I$, i.e. (50) and (66).

#### 3.3.1 Linearized gravity

The "equation of motion" (50) for $\omega^I_\mu$ was a simple consequence of the Bianchi identity $d\mathcal{F} = 0$ for the Kaluza-Klein vector $\mathcal{A}$ in (5). The linearized version of (50) was (51), i.e.:

$$\partial_i \omega^{I,i} = 0, \tag{58}$$

so $\omega^I_i$ is a divergenceless vector on $\mathcal{M}$ at the linearized level.

The linearized divergencelessness of $\omega^I_i$ can be used to define "multipole moments" of $\omega^I$ in a similar way that Geroch originally defined multipoles (in flat space) for a scalar field $\Phi$ satisfying the Laplace equation $D^2\Phi = 0$. For such a scalar, the multipole moments can be defined through integrals of the form [1]:

$$\int_K \left( \xi^{a_1} \cdots \xi^{a_\ell} D_m D_{a_1} \cdots D_{a_\ell} \Phi \right) dS^m, \tag{59}$$

where $\xi^a$ is a conformal Killing vector on $\mathcal{M}$, and $K$ is a two-sphere at infinity; the integral does not dependent of the precise choice of $K$ due to $D^2\Phi = 0$. In particular, if all possible integrals of the form (59) vanish, i.e. for all possible choices of conformal Killing vector $\xi$ and all possible integers $\ell \geq 0$, then all the multipole moments of $\Phi$ vanish — and in particular, $\Phi = 0$ is the only such solution to the Laplace equation.

In the case of $\omega^I_i$, we wish to demand that its multipole moments vanish, so that it does not contribute to the metric multipoles ultimately derived from the potential $\omega$. This is precisely the requirement of the vanishing of all possible integrals of the form:

$$\int_K \left( \xi^{a_1} \cdots \xi^{a_\ell} D_{a_1} \cdots D_{a_\ell} \omega^I_m \right) dS^m, \tag{60}$$

which does not depend on the precise choice of $K$ due to (58).

At the linearized gravity level, then, the "gauge" for $\omega^I$ must be such that (58) is satisfied, and such that $\omega^I$ has zero multipoles on the flat three-dimensional background, i.e. all integrals of the form (60) vanish. Note that this also precludes further "gauge" transformations $\omega^I \to \omega^I + 2dA^{(0)}$; from (58) it follows that $A^{(0)}$ must be harmonic, $D^2 A^{(0)} = 0$; and from the vanishing of (60) it follows that $A^{(0)} = 0$ is the only possibly solution.

**Einstein-Maxwell** It is interesting to apply the above reasoning on the Einstein-Maxwell improvement vector (32), i.e.:

$$\omega_\mu^I = -\frac{1}{2}\left(\tilde{\rho}\,\nabla_\mu\rho - \rho\,\nabla_\mu\tilde{\rho}\right). \tag{61}$$

At the linearized level, the equations of motion for $\rho, \tilde{\rho}$ are simply:

$$\partial^2\rho = \partial^2\tilde{\rho} = 0, \tag{62}$$

so that indeed:

$$2\partial_i(\omega^{I,i}) = -\tilde{\rho}\,\partial^2\rho + \rho\,\partial^2\tilde{\rho} = 0. \tag{63}$$

Moreover, since both $\rho$ and $\tilde{\rho}$ are harmonic functions, it follows that $\omega_i^I$ has vanishing multipoles; this is most easily seen using the expansion of e.g. $\rho$ using (constant) STF tensors $\rho_{A_\ell}$ as:

$$\rho = \sum_{\ell \geq 0} \rho_{A_\ell} \frac{N_{A_\ell}}{r^{l+1}}, \tag{64}$$

and then using properties of multiplying two such expansions as discussed in Appendix A. We conclude that (61) is indeed the unique possible Einstein-Maxell improvement vector under our "gauge fixing" conditions, to linearized order.[11]

### 3.3.2 Using ACMC coordinates

It is not obvious to generalize the above, coordinate-invariant discussion involving the equation of motion (50) to the non-linear level. However, fixing the gauge for $\boldsymbol{\omega}^I$ becomes very easy when we assume and use the existence of a harmonic ACMC coordinate system in which its expansion satisfies (20) (see Section 3.4 for a further discussion on this assumption).

In such an ACMC coordinate system, the twist vector $\omega_\mu$ will still be given by $\omega_0 = 0$ and (23), which can be slightly rewritten as:

$$\omega_i = -\sum_{\ell=1}^{\infty}\left(\frac{4\ell(2\ell-1)!!}{(\ell+1)!}S_{A_\ell}\partial_i\left[\frac{N_{A_\ell}}{r^{\ell+1}}\right] + \frac{\mathcal{S}_{\ell-1}}{r^{\ell+1}}\right). \tag{65}$$

The condition that $\omega_i^I$ does not contribute to the multipoles as derived from $\omega$, through $\nabla_\mu\omega = \omega_\mu + \omega_\mu^I$, is precisely the statement that $\omega_i^I$ will only contribute to the unimportant, lower-order angular dependences $\mathcal{S}_{\ell-1}$ at every order $r^{-(\ell+1)}$ in the asymptotic expansion, and will leave the multipole terms $\sim S_{A_\ell}$ unaltered, i.e. that we must have (with $\omega_t^I = 0$):

$$\omega_i^I = \sum_{\ell=1}^{\infty}\frac{\mathcal{S}_{\ell-1}}{r^{\ell+1}}. \tag{66}$$

Indeed, when (66) holds, the integrated expression (24) still holds for the twist vector:

$$\omega = -\sum_{\ell=1}^{\infty}\frac{1}{r^{\ell+1}}\left[\frac{4\ell(2\ell-1)!!}{(\ell+1)!}S_{A_\ell}N_{A_\ell} + \mathcal{S}_{\ell-1}\right]. \tag{67}$$

Using $\omega$ constructed in this way, the rest of the Gürsel proof of equivalence also immediately applies; the Geroch-Hansen and Thorne formalisms will give identical multipole moments as long as (66) holds (and as long as the metric is suitably smooth at infinity — see Section 3.4).

---

[11] It is interesting to note that (61) has been used in the literature [7–9, 39, 40] as "the correct improvement vector" without any reference to the conditions mentioned here (nor other conditions that would fix its gauge), even though there is then in principle no reason not to consider $\boldsymbol{\omega}^I$ "gauge"-shifted by an arbitrary gradient instead.

**Existence of (66)**   We can easily show the existence of a "gauge" for $\omega_i^I$ such that both $d\boldsymbol{\omega}^I = 2\boldsymbol{W}^{(2)}$ and (66) hold, using the properties of STF tensors listed in Appendix A. We start by noting that $\boldsymbol{\omega}^I = 2\boldsymbol{B}^{(1)}$, in an ACMC coordinate system, satisfies $B_0^{(1)} = 0$ and $\partial_t B_i^{(1)} = 0$; moreover, we have:

$$2\partial_{[i}B_{j]}^{(1)} = -\epsilon_{ijk0}g^{kl}R_{l0}\,, \tag{68}$$

where we used the definition of $\boldsymbol{B}^{(1)}$, the Einstein equations, and we ignored a term $\sim Rg_{i0}$ since this involves a product of (the non-constant part of) at least two metric tensors. Since derivatives can never introduce higher-order angular dependence (i.e. it can never convert a term $\mathcal{S}_{\ell-1}/r^{\ell+1}$ into $\mathcal{S}_{\ell'}/r^{\ell'+1}$), it suffices to show that:

$$R_{i0} = \sum_{\ell=1}^{\infty} \frac{\mathcal{S}_{\ell-1}}{r^{\ell+1}}\,, \tag{69}$$

in order to conclude that a "gauge" exists where (66) holds. With a few somewhat tedious calculations, and using that any product of two or more (non-constant parts of) metric tensors will automatically give subleading angular dependences that satisfy (69), we find:

$$R_{i0} = R^j{}_{ij0} = \partial_j \Gamma_{i0}^j + \mathbb{S} = \frac{1}{2}(\partial_i \partial_j g_{0j} - (\partial_j)^2 g_{0i}) + \mathbb{S}\,, \tag{70}$$

where we used the shorthand $\mathbb{S} = \sum_{\ell=1}^{\infty} \mathcal{S}_{\ell-1}/r^{\ell+1}$ to denote any subleading angular dependence. Now, we use the explicit ACMC expansion (20) to calculate:

$$\partial_i \partial_j g_{0j} - (\partial_j)^2 g_{0i} = -\sum_{\ell=1}^{\infty} \frac{4\ell(2\ell-1)!!}{(\ell+1)!} S_{kA_{\ell-1}} \left[\epsilon_{jka_\ell}\partial_j\partial_i\left(\frac{N_{A_\ell}}{r^{\ell+1}}\right) - \epsilon_{ika_\ell}\partial_j^2\left(\frac{N_{A_\ell}}{r^{\ell+1}}\right)\right] + \mathbb{S}\,. \tag{71}$$

Due to the derivatives, the highest possible angular dependence (at order $r^{-(\ell+3)}$) of the first term in the square brackets must be proportional to the tensor $[N_{A_\ell ij}]^{\text{STF}}$. This is symmetric in the indices $a_\ell$ and $j$, but is contracted by the antisymmetric tensor $\epsilon_{jka_\ell}$ — so we conclude this term must be $\sim \mathbb{S}$. The second term's highest angular dependence (at order $r^{-(\ell+3)}$) must be proportional to $[N_{A_\ell jj}]^{\text{STF}}$, but this vanishes due to the tensor's tracelessness; so also this term is $\sim \mathbb{S}$. We conclude that $R_{i0} = \mathbb{S}$, which is (69) as we needed to show.

**Satisfying the equation of motion with (66)**   Finally, we can also discuss the "equation of motion" (50) for $\omega_i^I$. The most general form of $\omega_i^I$ (i.e. not necessarily satisfying (66)) is:

$$\omega_i^I = \sum_{\ell=1}^{\infty}\left(\tilde{S}_{iA_{\ell-1}}\frac{N_{iA_{\ell-1}}}{r^{\ell+1}} + \frac{\mathcal{S}_{\ell-1}}{r^{\ell+1}}\right) = \sum_{\ell=1}^{\infty}\left(\tilde{S}_{iA_{\ell-1}}\partial_i\left(\frac{N_{A_{\ell-1}}}{r^\ell}\right) + \frac{\mathcal{S}_{\ell-1}}{r^{\ell+1}}\right)\,. \tag{72}$$

At the linear level (as discussed above), (50) becomes $\partial_i\omega^{I,i} = 0$ and so the coefficients $\tilde{S}_{A_\ell}$ are completely undetermined by this equation of motion; it follows that at the linear level it is certainly possible to choose a "gauge" for $\omega^I$ that satisfies (66). Note that all coefficients $\tilde{S}_{A_\ell}$ vanishing in (72) is of course precisely equivalent to the statement that all multipole integrals of the form (60) vanish, discussed above.

To generalize to non-linear order, we can use a similar argument as Thorne uses [4] (see Section IX and X on p. 327-332 therein) to construct the general non-linear ACMC expressions (20) from the linearized metric expressions $h_{\mu\nu}$ (which only features the multipole terms $\sim M_{A_\ell}, S_{A_\ell}$). In essence, Thorne shows, in constructing (20) from the linearized metric, that new leading order angular dependences (i.e. mixing with the multipole terms $\sim M_{A_\ell}, S_{A_\ell}$) will not appear at non-linear order as they could be re-interpreted as part of the *linearized* solution. Similarly, the only possible terms that appear in (72) order by order in $h$ are of the form $\sim \mathcal{S}_{\ell-1}/r^{\ell+1}$, since terms $\sim \tilde{S}_{A_\ell}$ could be reinterpreted as part of the linearized solution (and subsequently set to zero).

## 3.4 The existence of the ACMC expansion and smoothness of the metric

In the Geroch-Hansen formalism (see Section 2.1), a crucial ingredient was the existence of a conformal factor $\Omega$ and local coordinates $\hat{x}^i$ at the point $\Lambda$ at (compactified) infinity such that the metric coefficients were smooth when expressed in these coordinates. Gürsel's proof of the equivalence between the Geroch-Hansen and Thorne's ACMC formalisms (see Section 2.3) relies on the existence of *harmonic* ACMC coordinates and an appropriate choice of $\Omega$ and $\hat{x}^i$ which are guaranteed to make the metric coefficients smooth at $\Lambda$.

For vacuum spacetimes, Thorne showed that the general form of the linearized metric solution to Einstein's equations near infinity in the de Donder gauge is non-linearly completed in a way that satisfies the ACMC condition [4], so that de Donder coordinates are an explicit example of a possible harmonic ACMC coordinate system. Moreover, the metric coefficients $g_{\mu\nu}$ are analytic functions of the de Donder coordinates for Einstein metrics [41]. (Other relevant smoothness properties of vacuum spacetime metrics at infinity are given in [5, 12–14, 35, 36], as also cited above in Sections 2.1 and 2.3.)

Beyond vacuum spacetimes, the smoothness of the metric at infinity is not guaranteed, nor is the existence of a suitable ACMC coordinate system. In Sections 3.1-3.3, these were then implicit assumptions underlying the applicability of the Geroch-Hansen formalism for non-vacuum spacetimes and its equivalence to Thorne's ACMC formalism. Although these assumptions of smoothness at infinity and the existence of ACMC coordinates may seem like mild conditions on the metric, in this Section we will discuss how they could conceivably fail in a non-vacuum spacetime.

The existence of an ACMC coordinate system, and its relation to the canonical harmonic gauge of [11] is discussed in Sections 3.4.1 and 3.4.2. The smoothness of the metric and how this relates to the canonical harmonic gauge and ACMC expansions is then discussed in Sections 3.4.3 and 3.4.4. The tentative conclusion is that the ACMC formalism can be more general than the Geroch-Hansen formalism, since it is conceivable that metrics exist for which ACMC coordinates can be found but are not sufficiently smooth at infinity to allow for the Geroch-Hansen formalism to be applied.

### 3.4.1 Existence of ACMC coordinates

When is it possible to choose coordinates satisfying the ACMC condition (20) for a stationary, non-vacuum asymptotically flat metric? It is always possible for vacuum spacetimes, so intuitively it should also be possible for non-vacuum spacetimes as well, as long as the presence of matter "does not touch the structure at infinity too much". The discussion below in Section 3.4.2 gives an indication of making this notion more precise, and gives a simple counterexample by simply adding — by hand — a non-ACMC perturbation to flat space. However, it is important to stress that this example non-ACMC metric is *not* a solution to any known theory of gravity coupled to matter.[12]

Indeed, it seems that stationary solutions in most theories of (extended) gravity admit an ACMC coordinate system. This includes black holes and other stationary solutions in theories of (Einstein) gravity plus matter, such as the STU model discussed in Section 3.1.1 [17, 19]; black holes in higher-derivative-corrected gravity [22]; and black holes in scalar-tensor theories of gravity such as (Jordan-)Brans-Dicke [42–44].[13] In fact, I am only aware of one example

---

[12]In fact, the counterexample in Section 3.4.2 is very similar to the multipole situation of bumpy black holes, of which the multipoles were discussed in [30]; the bumpy deformations of Kerr are not a solution to any known theory of gravity coupled to matter, nor can they be brought to ACMC-*N* form for arbitrary high *N*, as mentioned in Section 1.1.

[13]Specifically, using a similar analysis as in [17, 19, 22], I have checked that the metric in eq. (2) (for $\Phi_0 = 1$, necessary for asymptotic flatness) of [43] (which is the metric found in [42]) as well as the asymptotically flat metric in eq. (0.39) of [44], can be brought to ACMC form. Note that for these stationary (J)DB solutions,

of a stationary metric in a theory where the ACMC expansion fails. This is the so-called "disformal Kerr" metric [45, 46], which is constructed from "stealth black hole" in a DHOST theory; the disformal Kerr metric is [45]:

$$
\tilde{ds}^2 = -\left(1 - \frac{2Mr}{\rho^2}\right)dt^2 - \frac{4\sqrt{1+D}Mar\sin^2\theta}{\rho^2}dtd\phi + \frac{\sin^2\theta}{\rho^2}\left[(r^2+a^2)^2 - a^2\Delta\sin^2\theta\right]d\phi^2
$$
$$
+ \frac{\rho^2\Delta - 2M(1+D)rD(a^2+r^2)}{\Delta^2}dr^2 - 2D\frac{\sqrt{2Mr(a^2+r^2)}}{\Delta}dtdr + \rho^2 d\theta^2 \,,
$$
$$
\rho^2 = r^2 + a^2\cos^2\theta \,, \qquad \Delta = r^2 + a^2 - 2M(1+D)r \,.
$$
(73)

Here, $D$ is a parameter indicating the "disformation" away from Kerr. It is easy to see that there is no possible coordinate transformation that will make this metric ACMC to all orders. In fact, for this metric, the metric components are no longer analytic functions of the coordinates (due to the factor $\sqrt{r}$ in $g_{tr}$), which is a necessary condition for an ACMC coordinate system to exist; see the discussion below in Sections 3.4.3 and 3.4.4.

### 3.4.2 The ACMC expansion and canonical harmonic gauge

In [11] (see section 2.3 therein), convenient expressions are given for the most general (not necessarily vacuum) linearized trace-reversed metric perturbation $\gamma_{\mu\nu} = \eta^{\mu\alpha}\eta^{\nu\beta}h_{\alpha\beta} - \frac{1}{2}\eta^{\mu\nu}\eta^{\alpha\beta}h_{\alpha\beta}$ (where the metric is linearized as $g_{\mu\nu} = \eta_{\mu\nu} + h_{\mu\nu} + \mathcal{O}(h^2)$) in what was called the "canonical harmonic gauge", where $\partial_\mu\gamma^{\mu\nu} = 0$:

$$
\begin{aligned}
\gamma_{00} &= \partial_{A_\ell}\mathcal{A}_{A_\ell} \,, \\
\gamma_{0i} &= \partial_{A_{\ell-1}}\mathcal{B}_{iA_{\ell-1}} + \partial_{pA_{\ell-1}}(\epsilon_{ipq}\mathcal{C}_{qA_{\ell-1}}) + \partial_{iA_\ell}\mathcal{D}_{A_\ell} \,, \\
\gamma_{ij} &= \delta_{ij}\partial_{A_\ell}\mathcal{E}_{A_\ell} + \partial_{A_{l-2}}\mathcal{F}_{ijA_{\ell-2}} + \partial_{pA_{\ell-2}}(\epsilon_{pq(i}\mathcal{G}_{j)A_{\ell-2}}) \\
&\quad + \left[\partial_{jA_{\ell-1}}\mathcal{H}_{iA_{\ell-1}} + \partial_{jpA_{\ell-1}}(\epsilon_{ipq}\mathcal{N}_{qA_{\ell-1}})\right]^{\text{STF}} + \partial_{ijA_\ell}\mathcal{K}_{A_\ell} \,.
\end{aligned}
$$
(74)

The functions $\mathcal{A}_{A_\ell}, \mathcal{B}_{A_\ell}, \mathcal{C}_{A_\ell}, \mathcal{D}_{A_\ell}, \mathcal{E}_{A_\ell}, \mathcal{F}_{A_\ell}, \mathcal{G}_{A_\ell}, \mathcal{H}_{A_\ell}, \mathcal{N}_{A_\ell}, \mathcal{K}_{A_\ell}$ are all STF tensors; the de Donder gauge further fixes $\mathcal{B}_{A_\ell}, \mathcal{E}_{A_\ell}, \mathcal{F}_{A_\ell}, \mathcal{G}_{A_\ell}$ in terms of the others.

The remaining functions $\mathcal{A}_{A_\ell}, \mathcal{C}_{A_\ell}, \mathcal{D}_{A_\ell}, \mathcal{H}_{A_\ell}, \mathcal{N}_{A_\ell}, \mathcal{K}_{A_\ell}$ are in general functions of the retarded time $u = t - r$ and $r$, and their form depends on the details of the solutions. For vacuum Einstein solutions, $\mathcal{D}_{A_\ell} = \mathcal{H}_{A_\ell} = \mathcal{N}_{A_\ell} = \mathcal{K}_{A_\ell} = 0$ [11], but for general non-vacuum solutions this need not be true.

The existence of an ACMC expansion for this stationary metric is equivalent with the condition that:

$$
\left\{\mathcal{A}_{A_\ell}, \mathcal{C}_{A_\ell}, \mathcal{D}_{A_\ell}, \mathcal{H}_{A_\ell}, \mathcal{N}_{A_\ell}, \mathcal{K}_{A_\ell}\right\} = \mathcal{O}\left(\frac{1}{r}\right) \,.
$$
(75)

Note that for a stationary metric, these functions can only be functions of $r$ and not of $u$.

It was implied in [11] that bringing the metric to this canonical harmonic gauge is equivalent to bringing it to a (harmonic) ACMC gauge. However, while true for vacuum solutions, this is not necessarily true for non-vacuum solutions. A simple counter-example is to take:

$$
\gamma_{00} = \frac{\cos^3\theta}{r^2} \,, \qquad \gamma_{0i} = \gamma_{ij} = 0
$$
(76)

(Note that this implies $h_{00} = \gamma_{00}/2$ and $h_{ij} = \delta_{ij}\gamma_{00}/2$). It is quite easy to see that the resulting metric $g_{\mu\nu}$ is not ACMC — indeed, it is *impossible* to find a coordinate transformation to bring the metric into an ACMC frame! This is then an example of a metric that does not

---

$R_{\mu\nu} \propto \partial_\mu\Phi\partial_\nu\Phi$ for the BD scalar $\Phi$ and so trivially $\boldsymbol{\omega}^I = W^{(2)} = 0$ since $\xi \cdot \partial\Phi = 0$.

admit an ACMC expansion. However, this metric is in canonical harmonic gauge. So the canonical harmonic gauge and the ACMC expansion are not equivalent: an ACMC coordinate system can fail to exist whereas the canonical harmonic gauge will always exist. It is important to supplement the canonical harmonic gauge with the condition (75) in order for it to be equivalent with the ACMC expansion — e.g. for (76), we have $\mathcal{A}_{ijk} \sim r$, which violates (75). The condition (75) somehow formalizes the idea of "not touching the structure at infinity" at all orders in $1/r$ that was mentioned above in Section 3.4.1.

Finally, from the above discussion, the condition (75) is clearly necessary and sufficient for the existence of an ACMC expansion. For stationary metrics, it seems this is an extra condition that needs to be imposed on the metric and in particular cannot be derived from asymptotic flatness. However, we note that, for *non*-stationary metrics, the condition (75) actually becomes equivalent to the condition of asymptotic flatness, since the spatial derivatives in (74) will also pick up contributions from the $u$-dependence of the STF tensors. This is a further indication of the naturalness of the condition (75). A final indication for the naturalness of this condition is given by demanding that the metric is smooth at infinity, discussed below.

### 3.4.3 Smoothness of the metric: Simple examples

The Geroch-Hansen formalism relies on the existence of a conformal factor $\Omega$ and local coordinates around the point at infinity $\Lambda$ such that the coefficients of the conformally compactified metric $\tilde{h}_{ab}$ as well as $\Omega$ are smooth at $\Lambda$. For vacuum spacetimes, a choice of such coordinates and $\Omega$ always exists [12–14,35,36]. However, for more general spacetimes, this existence can be a subtle matter.[14] To illustrate this, we consider a few simple (linearized) examples here.

Consider the linearized metric in canonical harmonic gauge (74) where only the tensors $\mathcal{A}_{A_\ell}$ are non-zero, so that only $\gamma_{00} \neq 0$ (and note that $h_{00} = \gamma_{00}/2$ and $h_{ij} = \delta_{ij}\gamma_{00}/2$). Following the Geroch-Hansen procedure to linear order in $h$, we have:

$$\Phi_M = \frac{1}{4}\gamma_{00} = \frac{1}{4}\partial_{A_\ell}\mathcal{A}_{A_\ell}\,, \tag{77}$$

and it is easy to see that the induced three-dimensional metric $h_{ab}$ is simply flat at the linearized level. We can choose new coordinates $\hat{x}^i = x^i/r^2$ and the conformal factor $\Omega = 1/r^2 = \hat{r}^2$ such that $\tilde{h}_{ij} = \Omega^2\delta_{ij}$ and $\tilde{h}_{\hat{i}\hat{j}} = \delta_{\hat{i}\hat{j}}$ — so the conformally compactified metric is again simply flat, and the compactified point at infinity $\Lambda$ is at the origin of the Cartesian $\hat{x}^i$ coordinates. In particular, all of the mass multipoles will simply be given by:

$$M_{A_\ell} = [\partial_{\hat{A}_\ell}\tilde{\Phi}_M]_{\hat{x}^i=0}^{\text{STF}}\,, \tag{78}$$

where the derivatives are with respect to the coordinates $\hat{x}^i$, and with $\tilde{\Phi}_M = \Omega^{-1/2}\Phi_M = r\gamma_{00}/4$.

In this setup, it is interesting to consider a few different choices for $\Phi_M$. For example, consider:

$$\Phi_M = c_0\frac{1}{r^3} + c_2\frac{P_2(\cos\theta)}{r^3}\,, \tag{79}$$

where $P_n$ are the Legendre polynomials. The compactified scalar $\tilde{\Phi}_M$, expressed in the coordinates $\hat{x}^i$, is given by:

$$\tilde{\Phi}_M = c_0(\hat{x}^2 + \hat{y}^2 + \hat{z}^2) - \frac{1}{2}c_2(\hat{x}^2 + \hat{y}^2 - 2\hat{z}^2)\,. \tag{80}$$

This is clearly smooth at $\Lambda$ (i.e. $\hat{x}^i = 0$). It results in a non-trivial mass quadrupole:

$$M_{xx} = M_{yy} = -c_2\,, \qquad M_{zz} = 2c_2\,. \tag{81}$$

---

[14]Of course, for specific theories, it can be possible to prove similar smoothness conditions on the metric as for vacuum spacetimes. For example, for stationary electrovacuum spacetimes, see [7].

The coefficient $c_0$ does not contribute to any multipoles.

On the other hand, we could instead take:

$$\Phi_M = c_1 \frac{\cos\theta}{r^3} + c_3 \frac{P_3(\cos\theta)}{r^3}, \tag{82}$$

which gives:

$$\tilde{\Phi}_M = c_1 \hat{z} \sqrt{\hat{x}^2 + \hat{y}^2 + \hat{z}^2} + \frac{c_3}{2} \hat{z} \frac{2\hat{z}^2 - 3\hat{x}^2 - 2\hat{y}^2}{\sqrt{\hat{x}^2 + \hat{y}^2 + \hat{z}^2}}. \tag{83}$$

This is not a smooth function at $\Lambda$; indeed, if one were to calculate $M_{ij}$ from (78), one would find that $\partial_i \partial_j \tilde{\Phi}_M$ is not continuous at $\Lambda$ — so $M_{ij}$ is not well-defined. In fact, this can be generalized to any higher-order angular dependence as well: if we take $\Phi_M \sim P_{\ell'}(\cos\theta)/r^\ell$, the resulting $\tilde{\Phi}_M \sim \hat{r}^{-n}$ (for $n > 0$), so that $\tilde{\Phi}_M$ is not smooth at $\Lambda$ (where $\hat{r} = 0$) and the multipoles are not well defined.

### 3.4.4 Smoothness of the metric: General considerations

From the above example, it seems clear that if an ACMC coordinate system does not exist, the Geroch-Hansen formalism will most likely also not be applicable. It will fail precisely because it will be impossible to find a conformal factor $\Omega$ and coordinates $\hat{x}^i$ such that $\Omega$ and the conformal metric $\tilde{h}_{ab}$ is smooth at $\Lambda$. In the language of the canonical harmonic gauge (74), it seems that the fall-off (75) is a necessary condition in order for the metric to be smooth at infinity. In the example (82) above with $c_3 \neq 0$, we had $\mathcal{A}_{ijk} \sim \log r$, which is certainly not smooth at $r \to \infty$.

However, the above example also shows us that the existence of an ACMC coordinate system is not sufficient for the Geroch-Hansen multipoles to be well-defined. The example (82) with $c_3 = 0$ but $c_1 \neq 0$ certainly satisfies the ACMC condition, but nevertheless it is clear that there is no choice of coordinates $\hat{x}^i$ and $\Omega$ that could make $\tilde{\Phi}_M$ smooth at $\Lambda$.

It is conceivable that a metric allows for an ACMC expansion without the Geroch-Hansen formalism being applicable (as the example (82) with $c_1 \neq 0$ shows). However, the converse is not true: if the ACMC formalism is not applicable, then the above arguments suggest the Geroch-Hansen formalism will fail as well, since the higher-order angular dependence in the non-ACMC metric leads to an unavoidable non-smoothness of the (compactified) metric at infinity. In this sense, then, the ACMC formalism can be more general than the Geroch-Hansen one.

It would certainly be interesting to find an example of a solution to a theory which admits an ACMC expansion but does not allow for the application of Geroch-Hansen; to the best of my knowledge no such solution is currently known.

## Acknowledgments

I wish to thank I. Bena, N. Bobev, P. Cano, G. Compère, K. Fransen, B. Ganchev, A. Puhm, and A. Ruipérez for discussions on multipoles, and especially I. Bena for prodding me about these issues of generalizing multipoles to arbitrary non-vacuum spacetimes.

**Funding information** I am supported by FWO Research Project G.0926.17N. This work is also partially supported by the KU Leuven C1 grant ZKD1118 C16/16/005.

## A  STF tensors

Symmetric and trace-free (STF) tensors and their properties are important in the multipole story. Using $n^i = x^i/r$ with $r = \sqrt{(x^1)^2 + (x^2)^2 + (x^3)^2}$, a first interesting property is:

$$\partial_{A_\ell}\left(\frac{1}{r}\right) = (-1)^\ell (2\ell - 1)!! \frac{[N_{A_\ell}]^{\text{STF}}}{r^{\ell+1}}. \tag{A.1}$$

We used $A_\ell = a_1 \cdots a_\ell$ as a shorthand over $\ell$ indices, so that in particular $\partial_{A_\ell} = \partial_{a_1} \cdots \partial_{a_\ell}$. The shorthand $N_{A_\ell} = n_{a_1} \cdots n_{a_\ell}$ denotes a purely angular dependence involving (up to) the order $\ell$ spherical harmonics. The superscript STF means to take the symmetric and trace-free part only. For example, $[n_i n_j]^{\text{STF}} = n_i n_j - \delta_{ij}/3$. See [4] for the more general formulae. This equation (A.1) is related to the general asymptotic expansion of a harmonic function $V$:

$$V = \sum_{\ell=0}^{\infty} \frac{(2\ell - 1)!!}{\ell!} \frac{M_{A_\ell} N_{A_\ell}}{r^{\ell+1}}, \tag{A.2}$$

where the multipole tensors $M_{A_\ell} = [M_{A_\ell}]^{\text{STF}}$ are all symmetric and trace-free. Note that there is a one-to-one correspondence between spherical harmonics of degree $\ell$ and STF tensors $[N_{A_\ell}]^{\text{STF}}$, which we will not need here (see e.g. [4]).

The formula (A.1) has important consequences. For example, it implies that a spatial derivative acting on a formula can never "increase" the angular dependence; schematically:

$$\partial_i\left(\frac{N_{A_\ell}}{r^{\ell+\ell'}}\right) \sim \frac{N_{A_\ell i}}{r^{\ell+\ell'+1}} + \frac{\mathcal{S}_{\ell-1}}{r^{\ell+\ell'+1}}, \tag{A.3}$$

where the $\sim$ denotes that the equation is schematic and does not take into account the constant of proportionality, and $\mathcal{S}_{\ell-1}$ denotes angular dependence up to at most order $\ell - 1$ spherical harmonics (so $N_{A_{\ell-1}}, N_{A_{\ell-2}}, \cdots$). This fact (A.3) is very important, as it implies that derivatives of the metric components in ACMC coordinates (20) will never "mix" the multipoles $M_{A_\ell}, S_{A_\ell}$ with the subleading angular parts $\mathcal{S}_{\ell-1}$ in the *leading* angular dependence at every order $\ell$ (i.e. $N_{A_\ell}/r^{\ell+1}$).

Finally, the product of two non-constant parts of metric tensors in ACMC coordinates always give a "subleading angular" part. In other words:

$$\left(\sum_{\ell=1}^{\infty} \frac{M_{A_\ell} N_{A_\ell} + \mathcal{S}_{\ell-1}}{r^{\ell+1}}\right)\left(\sum_{\ell=1}^{\infty} \frac{M'_{A_\ell} N_{A_\ell} + \mathcal{S}_{\ell-1}}{r^{\ell+1}}\right) = \sum_{\ell=1}^{\infty} \frac{\mathcal{S}_{\ell-1}}{r^{\ell+1}}. \tag{A.4}$$

This is important in the considerations of Section 3.3.2.[15]

## B  Improvement twist vector for $\mathcal{N} = 2$ supergravity

Here, I show that that the improvement twist vector (36) is indeed the correct improvement twist vector for the Lagrangian (35).[16] This means that, for (36) and (35), we have:

$$\partial_{[\mu}\omega^I_{\nu]} = +\epsilon_{\mu\nu\rho\sigma}\xi^\rho T^\sigma_{\ \lambda}\xi^\lambda. \tag{B.1}$$

The total energy momentum tensor can be written as the sum of a gauge field contribution and a scalar field contribution:

$$T_{\mu\nu} = T^{(F)}_{\mu\nu} + T^{(S)}_{\mu\nu} \tag{B.2}$$

---

[15]See [19] where similar arguments were important in calculating multipoles.

[16]Note that I follow almost the same normalizations and conventions of (A.16) in [19], except that I flip the sign of $I_{\Lambda\Sigma}$ so that $I_{\Lambda\Sigma} = \delta_{\Lambda\Sigma}$ is the vacuum flat space solution.

(The piece $T^{(F)}_{\mu\nu}$ is the piece that survives when the scalars are constants; $T^{(S)}_{\mu\nu}$ is obtained when all the gauge fields vanish).

The energy-momentum tensor of the scalar fields is:

$$T^{(S)}_{\mu\nu} = 4g_{IJ}\partial_\mu z^I \partial_\nu \bar{z}^J - 2g_{\mu\nu}g_{IJ}\partial_\rho z^I \partial^\rho \bar{z}^J \,. \tag{B.3}$$

The scalar field respecting the symmetry $\xi$ means that $\mathcal{L}_\xi z^I = \xi^\mu \partial_\mu z^I = 0$, and similarly for $\bar{z}^I$. In this case, we can calculate:

$$\xi^{[\alpha}T^{(S)\beta]}{}_\gamma \,\xi^\gamma = 4g_{IJ}\xi^{[\alpha}\partial^{\beta]}z^I \partial_\gamma \bar{z}^J \xi^\gamma = 0\,, \tag{B.4}$$

where we note that the term in $T_{\mu\nu}$ that is $\propto g_{\mu\nu}$ does not contribute and we used $\xi\cdot\partial\bar{z}^I = 0$. From (B.4), it follows that the scalars do not contribute to (B.1) — this is entirely analogous to the case of a single, free scalar [11].

Turning to the gauge fields, we first define the dual field strength $G_\Lambda$:

$$G_{\Lambda\mu\nu} \equiv -2\frac{\delta\mathcal{L}}{\delta F^{\Lambda\mu\nu}} = I_{\Lambda\Sigma}F^\Sigma_{\mu\nu} - R_{\Lambda\Sigma}\tilde{F}^\Sigma_{\mu\nu}\,. \tag{B.5}$$

The equations of motion for the gauge fields are then simply:

$$\nabla_\mu G_\Lambda^{\mu\nu} = 0\,. \tag{B.6}$$

Similarly to in pure Maxwell theory, the forms $F^\Lambda$ satisfy $dF^\Lambda = 0$ due to the Bianchi identities. The gauge field equations of motion are $d*G_\Lambda = 0$. Defining the Hodge dual $\tilde{G}_\Lambda$ of $G_\Lambda$ in the usual way, $\tilde{G}_\Lambda = *G_\Lambda$, it follows that we can always define a dual potential $\tilde{A}_\Lambda$ such that $\tilde{G}_\Lambda = d\tilde{A}_\Lambda$.

The Lagrangian can be rewritten in the useful form:

$$\mathcal{L} = R - 2g_{IJ}\partial_\mu z^I \partial^\mu \bar{z}^J - \frac{1}{4}F^\Lambda_{\mu\nu}G_\Lambda^{\mu\nu}\,. \tag{B.7}$$

From here, it is easy to see that we can write the energy-momentum tensor of the Maxwell fields as:

$$T^{(F)}_{\mu\nu} = \frac{1}{2}\left(F^\Lambda{}_{\mu\rho}\,G_{\Lambda\nu}{}^\rho - \frac{1}{4}g_{\mu\nu}F^\Lambda{}_{\rho\sigma}\,G_\Lambda{}^{\rho\sigma}\right)\,. \tag{B.8}$$

Note that $T^{(F)} = T^{(F)\mu}{}_\mu = 0$. Also, note that the expression $F^\Lambda{}_{\mu\rho}\,G_{\Lambda\nu}{}^\rho$ is automatically symmetric in the indices $(\mu\nu)$, which can be seen from using the definition of $G_\Lambda$ above and using the Schouten identity on the term involving $\tilde{F}$.

Having introduced $G_\Lambda$ in this way, it is clear that the calculation of the improvement twist vector $\omega^I$ is now analogous to the case of pure Maxwell theory. From (B.8), $\mathcal{L}_\xi A^\Lambda = \mathcal{L}_\xi \tilde{A}_\Lambda = 0$, and defining $\rho^\Lambda = \xi\cdot A^\Lambda$ and $\tilde{\rho}_\Lambda = \xi\cdot\tilde{A}_\Lambda$, it is a straightforward calculation to show that (36) indeed satisfies (B.1) for $T_{\mu\nu} = T^{(F)}_{\mu\nu}$.

## B.1 Gauge potentials, field strengths, and symmetries

It is straightforward to show that a gauge field $F = dA$ enjoys a Killing symmetry $\xi$, so $\mathcal{L}_\xi F = 0$, if and only if it is possible to choose its gauge potential $A$ to enjoy the same symmetry, so $\mathcal{L}_\xi A = 0$.

If $\mathcal{L}_\xi A = 0$, and using $\mathcal{L}_\xi = di_\xi + i_\xi d$, we have $i_\xi F = -di_\xi A$. Then $di_\xi F = 0$ and moreover $dF = 0$ from the Bianchi identity, so that $\mathcal{L}_\xi F = di_\xi F + i_\xi dF = 0$.

To show the converse, say we have $F$ such that $\mathcal{L}_\xi F = 0$, and a gauge potential $A$ such that $F = dA$. Then, using the Bianchi identity, we have $di_\xi F = 0$ so that $i_\xi F = d\rho$ for some potential $\rho$. Allowing for a gauge transformation, $A \rightarrow A' = A + d\Lambda$, we have:

$$\mathcal{L}_\xi A' = di_\xi A + i_\xi F + di_\xi d\Lambda = d(i_\xi A + \rho + i_\xi d\Lambda)\,. \tag{B.9}$$

From this, it is clear we can always find a gauge transformation parameter $\Lambda$ such that $\mathcal{L}_\xi A' = 0$. This is perhaps easiest to see in a coordinate system where $\xi = \partial_x$ for some coordinate $x$, in which we can write the equation:

$$i_\xi \mathbf{A} + \rho + i_\xi d\Lambda = A_x + \rho + \partial_x \Lambda = c,\qquad\text{(B.10)}$$

for some constant $c$; this equation can always be integrated for $\Lambda$ so that $\mathcal{L}_\xi A' = 0$ is satisfied.

### B.2 Uniqueness of electrostatic potentials

Even though $\mathbf{A}$ is not gauge invariant — even after demanding $\mathcal{L}_\xi \mathbf{A} = 0$ —, it is easy to see that the electrostatic potential $\rho = \xi \cdot A$ *is* unique when preserving $\mathcal{L}_\xi \mathbf{A} = 0$. Indeed, consider a gauge transformation $\mathbf{A} \to \mathbf{A}' = \mathbf{A} + d\alpha$ that preserves stationarity of the gauge field, $\mathcal{L}_\xi \mathbf{A} = \mathcal{L}_\xi \mathbf{A}' = 0$. Under this gauge transformation, $\rho \to \rho + \xi \cdot d\alpha = \rho + i_\xi d\alpha$. But from $\mathcal{L}_\xi \mathbf{A}' = 0$, it follow that $0 = \mathcal{L}_\xi d\alpha = d i_\xi d\alpha$, so that $i_\xi d\alpha = cte$. Moreover, this constant $cte$ must vanish at infinity (and thus everywhere).[17] We conclude that $\rho$ is invariant under gauge transformations that preserve stationarity of the gauge field $\mathcal{L}_\xi \mathbf{A} = 0$. Of course, an entirely analogous reasoning applies to show that the dual potential $\tilde{\rho}$ is invariant under gauge transformations that preserve stationarity of the dual gauge field, $\mathcal{L}_\xi \tilde{\mathbf{A}} = 0$.

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
