# Peer review of "Gravitational Multipoles in General Stationary Spacetimes"

_SciPost Physics, doi:SciPost Phys. 15, 154 (2023)_

## Round 1 · Referee Report · Anonymous (Referee 1) · 2023-6-13

Report

The author now addresses comments raised in my previous report. I think the paper is now suitable for publication. However, I have additional comments that might be helpful in improving the presentation of the paper, but I leave it to the author to decide whether and how to implement these comments. I also think that new additions to the paper, including the field equations for the mass and spin potentials, and the relation to Kaluza Klein reduction are very interesting.

Attachment

  • validity: -
  • significance: -
  • originality: -
  • clarity: -
  • formatting: -
  • grammar: -

Author:  Daniel Mayerson  on 2023-06-22  [id 3749]

(in reply to Report 1 on 2023-06-13)

I thank the referee for his remarks. I would be happy to adjust the wording of the abstract and the opening of section 3.1.2 to accommodate the remarks 2 and 3.

With respect to remark 1, it would certainly be nice to find a similar and less technically obscure gauge-fixing condition on the improvement form compared to the one given in the paper. However, unfortunately, I don't think the referee's suggestion is applicable.
The manifold in question does not have a boundary (or, at most, it has only a boundary at small r, but is unbounded for large r), so the formalism of 1904.12869, and in particular, the unique (Helmholtz) decomposition (2) in the referee report, does not hold. For example, in a standard counterexample to the uniqueness of the Helmholtz decomposition in non-compact spaces without boundaries, one can shift psi -> psi' = psi + H and phi->phi'=phi-H, where H is harmonic (d*dH = 0), and it can be seen that the decomposition (2) also holds with \delta psi' = 0 and orthogonality of psi' and phi'. (Such harmonic functions also clearly exist and can be non-trivial, eg. 1/r or its derivatives.)
Similarly, the demand that the improvement form would be orthogonal to (all) exact forms is not enough to fix the gauge conditions as given in the paper draft, as shifting such an improvement form by dH (with H harmonic) would still be allowed.
Incidentally, it is tempting to attempt a similar reasoning using a Helmholtz/Hodge decomposition of the improvement form but instead on the compactified manifold \tilde M, since on this compact manifold such Hodge decompositions are unique. However, to be able to do this would first require knowledge of how the improvement one-form transforms under conformal transformations. I strongly suspect that (again, unfortunately) the improvement one form does not transform in a simple way under such conformal transformations. (At least, there is no particular reason for it to transform simply.)

---

## Round 1 · Author Response

Full resubmission letter, reply to referees, and list of changes is given in pdf file.

---

## Editorial Decision

published